# Infusing Synthetic Data with Real-World Patterns for Zero-Shot Material State Segmentation

**Sagi Eppel [1,2*], Jolina Yining Li [2*], Manuel S. Drehwald[2], Alan Aspuru-Guzik[1,2]**
[1]Vector institute, [2]University of Toronto, * Equal contributions

## Abstract

Visual recognition of materials and their states is essential for understanding the physical world, from identifying wet regions on surfaces or stains on fabrics to detecting infected areas on plants or minerals in rocks. Collecting data that captures this vast variability is complex due to the scattered and gradual nature of material states. Manually annotating real-world images is constrained by cost and precision, while synthetic data, although accurate and inexpensive, lacks real-world diversity. This work aims to bridge this gap by infusing patterns automatically extracted from real-world images into synthetic data. Hence, patterns collected from natural images are used to generate and map materials into synthetic scenes. This unsupervised approach captures the complexity of the real world while maintaining the precision and scalability of synthetic data. We also present the first comprehensive benchmark for zero-shot material state segmentation, utilizing real-world images across a diverse range of domains, including food, soils, construction, plants, liquids, and more, each appears in various states such as wet, dry, infected, cooked, burned, and many others. The annotation includes partial similarity between regions with similar but not identical materials and hard segmentation of only identical material states. This benchmark eluded top foundation models, exposing the limitations of existing data collection methods. Meanwhile, nets trained on the infused data performed significantly better on this and related tasks. The dataset, code, and trained model are available at these URLs: 1, 2, 3, 4. We also share 300,000 extracted textures and SVBRDF/PBR materials to facilitate future datasets generation at these URLs: 1,2, 3, 4.

## 1 Introduction

Materials and their states form a vast array of patterns and textures that define the physical and visual world. Minerals in rocks, sediment in soil, dust on surfaces, infection on leaves, stains on fabrics, and foam in liquids are some of these almost infinite numbers of states and patterns. Segmenting these states in images is fundamental to the comprehension of the world and is essential for a wide range of tasks, from cooking and cleaning to construction and laboratory work. Currently, there is no comprehensive dataset or benchmark that addresses this general task, and even the top foundation models perform poorly on this problem (Section 6). This work presents the first general benchmark and datasets focused on this task, encompassing a wide range of material states and domains without being limited to specific materials or settings[7, 59]. This includes dealing with gradual transitions and partial similarity between states, as well as scattered shapes and fuzzy boundaries (Figure 1 top). For the benchmark, the goal is to achieve this using real-world images taken in the wild. The goal of the training set is to provide unlimited synthetic data that capture the complexity and variability of the real world while not being limited to a set of procedural rules and assets. Manual annotation of material states is highly complex from a human perspective, especially for cases with soft boundaries and gradual transitions or sparse and scattered patterns. In specific settings, it is sometimes possible

38th Conference on Neural Information Processing Systems (NeurIPS 2024) Track on Datasets and Benchmarks.

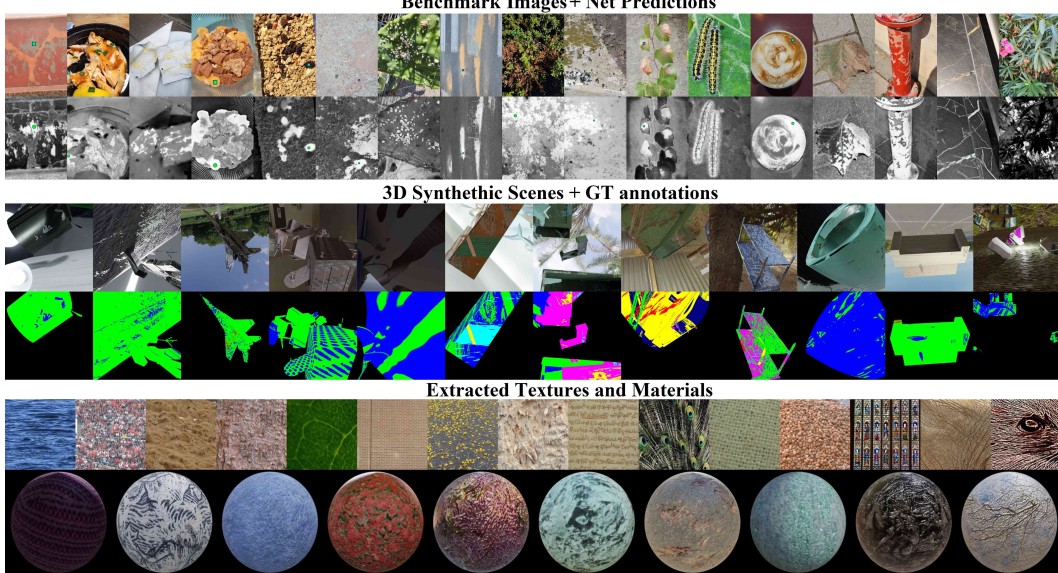

Figure 1: **Top: MatSeg benchmark images** and MatSeg trained net predictions. Predictions are shown as a material similarity map relative to a selected point in the image (marked green). **Center: Synthetic 3D scenes** and their annotations. Different colors in the annotation stand for different materials, black is the background. **Bottom: Textures and SVBRDF/PBR materials** automatically extracted from natural images. More samples can be seen in figs. 9, 12 and 13.

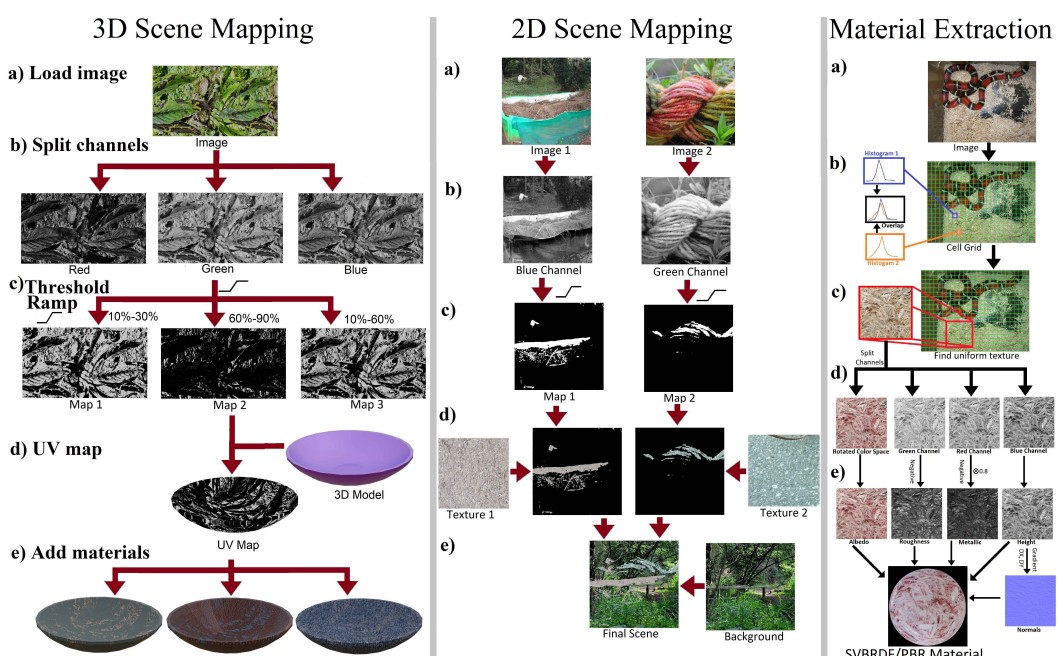

Figure 2: Extracting textures and patterns from real-world images and using them in synthetic scenes: **3D/2D Scene Mapping:** Mapping materials into synthetic scenes using maps extracted from natural images: a) Select a random image. b) Split the image into channels (R,G, B, H,S,V) and select one random channel. c) Apply a random ramp threshold to the selected channel to get a soft binary map. d) Use the map to position material on objects and scenes.
**Material Extraction:** a) Pick an image. b) Divide the image into a grid. For every grid cell, extract the distribution of colors and gradients. c) Identify a region for which all cells have similar distributions as a uniform texture. d,e) Pick random channels from the extracted texture image, augment them, and use the resulting maps as property maps (roughness, metallic, height, etc.) for the SVBRDF/PBR material.

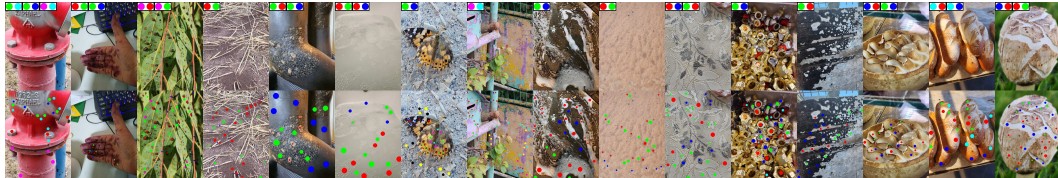

Figure 3: MatSeg benchmark with points-based annotation. Points of the same color are of the exact same material. Similar materials are marked as two or more dots of different colors in a box (top left). More samples can be found in fig. 11.

to find some method or additive to simplify the annotations. However, the unstructured and wide range of domains in the benchmark makes this unattainable in this case. Synthetic data can simulate soft and scattered states with perfect annotation. However, it fails to capture the vast complexity of material forms in the real world, and while it is possible to accurately represent specific domains using simulation or procedural rules[45]. Achieving this for the full range of material states in the world encompasses the entire material science and is unattainable.

## 1.1 Patterns infusion: Extracting and implanting real-world patterns in synthetic data

This work offers a method to bridge this gap between synthetic and real data by automatically extracting patterns and textures from real-world images and using these to generate and map materials into synthetic scenes (Figure 2). This approach combines the precision and scale of synthetic data with the vast variability of patterns in the real world. The hypothesis is that simple image properties like value, hue, saturation, or one of the Red, Green, or Blue channels (Figure 2) are often correlated with material regions or properties within a material texture[15, 63, 27]. For example, the dry part of a leaf, stains on fabrics, or pollution in water will often be darker or brighter than their surroundings. In these cases, the image brightness map will capture the material region. Similarly, within a single material texture, regions with higher reflectivity, transparency, or roughness will often be darker or brighter, allowing a single image channel to capture the property distribution. Assuming this is true for some fraction of cases and that important material states appear in a wide range of settings, extracting these maps from a broad range of images will capture these patterns. Once captured, these patterns can be used in various synthetic scenes where the original correlation is not applied and therefore will be learned by the net in general form (Figure 2). Thus, patterns are automatically extracted in simple cases with a strong correlation between material and a simple image property but are then rendered and learned in other scenes where this correlation does not occur. Since this is an unsupervised process, it will capture a significant amount of noise and unrelated patterns. However, neural nets are very effective in learning from noisy data, as demonstrated by methods such as CLIP and GPT[52, 44, 9]. Conversely, neural nets tend to perform poorly when trained on clean data with a narrow domain and then are expected to generalize to broader tasks[44, 29]. The fact that the net trained on this infused data outperforms top foundation models (SAM, Materialistic[34, 53]) on a wide range of benchmarks strongly supports this hypothesis. Another important aspect is the use of physics-based rendering (PBR). While the unsupervised method may extract patterns unrelated to materials (e.g., shadows), mapping these patterns onto 3D scenes using the Blender physics engine results in materials with realistic visual properties. For instance, if the unsupervised method extracts a pattern from a shadow in the image, this pattern will be used in the 3D-rendered scene to map physics-based materials onto object surfaces. Consequently, some materials in the scene may appear in shapes not derived from real-world materials, but their appearance and physical properties will be grounded in physics. We share the dataset and an a giant repository of extracted textures and SVBRDF/PBR materials collected by this method to enable the generation of future datasets (appendix A).

## 1.2 Zeros-Shot Material State Segmentation Benchmark

We present the first general benchmark for zero-shot material state segmentation. Benchmarks for specific material states like corrosion or dust have been published[48, 18]; however, no general zero-shot benchmark for material states that encompasses a wide range of fields without being limited to a specific set of states has been published so far. The benchmark contains 1,220 real-world

images with a wide range of materials and domains, such as food states (cooked / burned, etc.), plants (infected / dry), rocks / soil (minerals / sediment), construction / metals (rusted/worn), liquids (foam/precipitate), and many other states in a class-agnostic manner (figs. 1 and 3). The goal is to evaluate the segmentation of material states without being limited to a specific class, domain, or setting. This includes materials with complex or scattered shapes, fuzzy boundaries, or gradual transitions. This makes manual polygon-based annotation unfeasible. To solve this, we use point-based and similarity-based annotations. Hence, for each image, we sample multiple points that represent the distribution of each material state (fig. 3). Points with the exact same material state are grouped under the same label. We also define groups (materials) that have partial similarity. For example, points in group A are more similar to points in group B than to points in group C if materials A and B are similar but not identical. This captures the complexity of gradual transitions while also dealing with scattered shapes without needing to manually annotate the full image, which in many cases is unfeasible. The top image segmentation foundation models (SAM, Materialistic[34, 53]) perform poorly on this benchmark. This shows that the main training data generation approaches have a major gap in handling the material world.

### 1.3 Applications of zero-shot material segmentation

Existing methods and datasets for material segmentation are usually focused on specific domains and classes and rarely address scattered and soft boundaries. The MatSeg dataset does not aim to compete with domain-specific datasets but provides a more general solution where domain-specific training is not viable. This could be open-world problems like autonomous chemistry and material research or any wide-scope segmentation task that cannot be restricted to a predefined set of classes. In addition, the high generality can make it a good starting point for tasks with limited training data.

### 1.4 Summary of main contributions

1) Constructing the first general material state segmentation benchmark and dataset. This enables, for the first time, training and evaluation on this fundamental task and exposes a major gap in the capabilities of top image segmentation models.
2) Demonstrating an unsupervised method for extracting patterns and textures from real-world images and infusing them into synthetic scenes. This method enables the creation of synthetic data that automatically encompasses much of the complexity and diversity of the real world. The net trained on this data outperforms the top methods in various zero-shot benchmarks.
3) In addition to the dataset and the net, we also share a giant repository of extracted textures and materials to enable future dataset generation (appendix A).

## 2 Related works

**Unsupervised texture segmentation and extraction:** Methods for unsupervised texture extraction and image segmentation based on simple image features (color, shades, gradients) or statistical features have been extensively explored in the last 50 years[57, 63], but were mostly discarded in favor of deep learning approaches. These methods usually involve finding connected regions with uniform color and shades [15, 63, 27], or splitting the image into a grid and finding cells with a similar statistical distribution[57, 12, 42, 31]. The goal of this work is not to improve upon these methods, but rather to show that the pattern extracted by them while noisy and imprecise captures a broad range of features that are missing from leading data generation methods. Hence, these "ancient" methods offer a way to generate training data that can outperform modern data generation methods in important domains.
**Synthetic data and the domain gap:** Synthetic data and CGI images are commonly used to train machine learning models for computer vision tasks[36, 41, 56, 55, 47, 36]. Similarly to games and movies, this data is crafted using human-made assets[1, 2, 64, 3, 11, 19], scans, simulations, and procedural generation[45]. However, they often fail to capture the diversity and complexity of the real world, leading to a domain gap. To address this, domain adaptation techniques such as GANs are employed to make images more photorealistic or adapt them to different conditions[33, 65, 51, 46]. These methods mainly adjust the images without changing the underlying scene. Additionally, some approaches use procedural rules and mathematical functions to better mimic the world, these methods generate a wide range of patterns but are still limited by the generation rules[45, 10].

**Materials in synthetic scenes, representation, extraction and mapping:** Materials in CGI and synthetic scenes are usually represented as SV-BRDF or PBR materials[6, 43]. The term PBR is more common and will be used in this work. This representation is based on a few properties of the material surface, such as albedo, reflectance, normals, roughness, and transparency. Each property is represented as a 2D map that defines the value of this property on the surface. These maps are wrapped around the object to give it a material appearance. Generating these maps is done mostly manually by CGI artists and scans[1, 2, 64, 3]. Currently, there are a few thousand PBRs that are publicly available in open repositories[1, 2, 64, 3, 21, 61]. Recently, neural nets that generate PBR materials from input text or images have been suggested and embedded in various products[8, 30, 25, 20, 37, 14, 32, 58]. However, since these methods are trained using existing PBR material repositories, it is unclear if they can generate beyond the distribution on which they were trained. As far as we know, these works focus on generating assets for CGI artists and were not tested for generating training data machine learning methods.

**Zero-Shot and Class Agnostic Segmentation:** Methods for identifying and segmenting materials and textures without being limited to a specific set of classes or properties are often called one-shot, zero-shot, or class-agnostic. Zero-shot segmentation usually receives a query or a point in the image as input and outputs the region of the image corresponding to the query[60, 54, 13, 22, 34, 4, 53] Zero-Shot methods for texture segmentation were trained using synthetic data by projecting multiple material textures into an image and generating images with known segmentation maps, which are used to train nets [60, 54, 13, 53]. These methods gave good results for simple, class-independent, materials and texture segmentation. However, the distribution of materials it generates is either random or depends on a set of limited pre-made assets. Both cases don't aim to replicate the complexity of materials in the world, and up to this work, were not tested on material state segmentation.

# 3 Data generation: patterns extraction and infusion

The data infusion process described in fig. 1 and section 1.1 serves two main purposes. The first is to map existing materials into synthetic scenes (section 3.1). The second application involves extracting textures and PBR/SVBRDF materials from images (section 3.2). Both applications rely on the assumptions outlined in section 1.1 and are detailed below.

## 3.1 Unsupervised extraction of maps from images

Maps were extracted from images by randomly selecting a single image property such as saturation, hue, brightness, or one of the RGB channels (fig. 2b). Each of these features transforms the image into a 2D topological map. The resulting map is then processed using a fuzzy thresholding technique (color ramp, fig. 2c), which involves choosing two random threshold values. All elements of the map above the higher threshold are set to 1, those below the lower threshold are set to 0, and values in between are linearly interpolated between 0 and 1 (fig. 2d). This procedure creates maps with distinct segments while allowing for smooth and gradual transitions at boundaries. Because the images, channels, and threshold values are chosen randomly, the patterns extracted are unpredictable. However, visual inspection of these maps shows that they capture a diverse range of interesting and complex image patterns (fig. 4). Repeating this process several times can create maps with multiple distinct regions. These maps are used to map materials onto the surface of objects and scenes (fig. 4).

**Mapping and blending materials:** The values of cells in the generated maps serve as blending weights that dictate how materials are mapped into the scene (fig. 2d). Each point on the map gives the ratio in which different materials are combined at this point. In 2D applications, this map is used as the alpha channel, indicating the blending proportions of each material at each pixel. For 3D scenes, it is used as the UV map to control how materials are distributed across object surfaces (fig. 2d). This method captures not only shapes and patterns but also boundary and gradient profiles that determine how materials transition and merge into one another and the surrounding scene. As a result, it produces both gradual and sharp boundaries, depending on the gradient characteristics of the source image from which the patterns are derived.

**Creating 2D Scenes:** The simple way to use the extracted map is to define every map region as a different material in an image and map different textures into different regions (Figure 2.center.d).

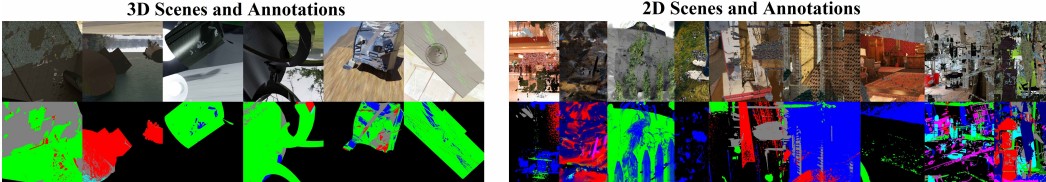

Figure 4: Examples of 2D and 3D synthetic scenes (top) and their annotations (bottom). In the annotation, each material is marked as a different color. A mixture of materials is marked as a weighted mixture of their colors. The background is marked black. More samples can be seen at fig. 13.

The resulting image is then overlaid on top of a random background image (Figure 2.center.e). Shadow-like effects can be achieved by using another extracted soft map to darken some regions according to the map value. In regions where the value of the map is between 0 and 1, several textures are mixed according to the values of the map (fig. 4).

**Creating 3D scenes:** A more advanced approach is to use the generated maps to set PBR materials on object surfaces in 3D scenes using CGI software such as Blender[8]. This is done by creating 3D scenes and wrapping the extracted maps around objects' surfaces in a process called UV mapping. Random PBR materials are then placed in each segment of the map (fig. 2.left.d,e). Background, random objects, ground, and illumination are added to generate full 3D scenes with physics-based rendered materials (PBR), natural illumination, shadows, and curved surfaces (fig. 1.center).

## 3.2 Extracting Textures and Materials

Materials in synthetic datasets are usually represented as SVBRDF/PBR, in which the spatial distribution of each material property (reflectivity, roughness, transparency, etc.) is given as a 2D map[6, 43]. Extracting textures and PBR materials from images involves first identifying regions with a uniform texture (which implies a uniform material, fig. 2.right). This texture image is then used to extract maps corresponding to each material's property.

### 3.2.1 Unsupervised texture extraction

Unsupervised texture extraction is traditionally done by finding image regions with a uniform distribution of color and gradients[57, 12, 42, 31, 62]. Finding these regions is done by splitting the image into square cells of 40x40 (or 75x75) pixels each and identifying square regions where the distribution of the RGB values and their gradients is similar for all the cells in the region. Regions of more than 6x6 (or 7x7) cells with similar distributions between all cells were extracted (fig. 2.right). More accurately, a region was considered to have uniform texture if the similarity of the distributions between each pair of cells in the region had a Jensen-Shannon distance of less than 0.5. This similarity was tested separately for each of the RGB channels and their gradients. Finally, regions with too-uniform values or very high or low values are ignored, as they only contain smooth, textureless regions. It's clear that this approach will fail to capture textures on bent surfaces or non-uniform illumination and shadows. However, the assumption is that for a large enough set of images, common material textures will appear in some cases in a simple setting and will be captured (section 1.1). Once captured, they will be projected into scenes with different illuminations, and bent surfaces and will be learned for the general case. Some textures extracted using this approach are shown in figs. 1 and 10. For more detail on this process see appendix I.

### 3.2.2 Generating material property maps from texture images

Extracting PBR material maps from RGB textures requires guessing the physical properties of the material at each point of the texture image. There is no straightforward way to achieve this using an RGB image. Once again, the assumption will be that the properties of the material (reflectivity, roughness, and transparency) are correlated, in some cases, with simple image properties like color (section 1.1), and for a common materials and a large enough set of images such correlation is bound to occur in some setting. Each material property map is generated by randomly choosing one of the six image channels (RGB or HSV), to represent this map (fig. 2.right.d), image augmentation,

thresholding, and random scaling/shifting were also used. Normal maps were generated by taking the gradient of the height/bumpiness map (fig. 2.right.e). In some cases, the map was randomly chosen to have a uniform random value. Some results are shown in figs. 1 and 9. In addition, materials were also generated by mixing maps of existing PBR materials as described in appendix F.

# 4 Material State Segmentation Benchmark

Establishing a benchmark for testing general material state segmentation beyond a specific set of materials and domains is essential for evaluating methods in this field. Such benchmarks must be based on real-world images and encompass a wide range of material states, environments, and domains. In addition, it needs to capture scattered and sparse shapes commonly formed by materials. Finally, to address mixtures and gradual transitions, the benchmark must capture partial similarity between materials. Currently, no benchmark is available that addresses these challenges.

**Image collection and annotation:** 1220 images for the benchmark were manually taken from a wide variety of real-world scenes (fig. 3). Including food at different levels of cooking, liquids, solutes, foams, plants, rocks, minerals, sediments, grounds, landscapes, construction sites, metals, relics, fabrics, labs and processes like cooking, wetting, rotting, infections, corrosion, stains, and many others. The goal is to capture as much of the world as possible, both in terms of materials and environments. For materials with scatter, soft boundaries, and gradual transitions, it is often impossible to segment the entire region belonging to a material state. An alternative approach is to sample points in the image that represent the main materials' states and their distribution. This allows annotators to focus on regions with clear annotation, but also to sample the harder and more complex regions without dealing with areas where the segmentation is unclear. The annotation procedure is as follows: 1) Pick points in each image that represent the distribution of each material state (fig. 3). 2) Group the points according to their material state such that all points belonging to the exact same material state will have the same label or group (points of the same color in fig. 3 are in the same group). 3) Assign relative similarity between groups. For example, if material B is a transition state between materials A and C, we can define the points in group B to be partially similar to both groups A and C, while groups A and C are not similar to each other. This circumvents the almost impossible task of assigning numerical similarity to material states (fig. 3). For more details on the benchmark points selection process see appendix H.

## 4.1 Evaluation metric

The evaluation approach is based on assessing the relative similarity between the points in the image (Figure 5.right). Every segmentation approach, soft or hard, can be easily converted to predict the similarity between two points in the image. For hard segmentation, this similarity will be one if the two points are in the same segment and zero if not. For evaluation, we use the triplet metric which goes as follows: a) Select three points in the image and define one as an anchor. b) Ask which of the remaining two points is more similar to the anchor according to the ground truth (GT) and according to the prediction; if the predictions and GT agree, we assign this as correct. c) If the two points have identical similarity to the anchor (according to the GT), we ignore this triplet. This is done for all sets of GT triplets in the image, and the average per material is used. For this metric, a 50% score means a random guess and a 100% score means perfect segmentation. The results are given in table 1.

## 4.2 Additional benchmarks

While there are no other benchmarks for general zero-shot material state segmentation, there are a number of benchmarks that deal with subdomains of this task. These include segmentation of specific material states, like corrosion[48], minerals/ores[17, 16, 26, 28], dust[18], leaf disease[5], soil types and states[49, 50], tissue microscopy[39, 40], or material phases (liquids, solids, foams, or powders)[24, 23]. Although each of these benchmarks is relatively narrow, when combined, they offer a way to cross-check the results of the main benchmark. Materialistic is a benchmark[53] that contains 50 images with different materials, not limited to specific classes. With materials segments that cover the whole object or regions with straight, smooth boundaries. It consists of distinct material types with no states or gradual transitions. Comparing the results of this benchmark to MatSeg offers a way to examine how much these elements affect segmentation accuracy. To evaluate the nets on these benchmarks, we select a random point in each material segment and use the nets to predict the

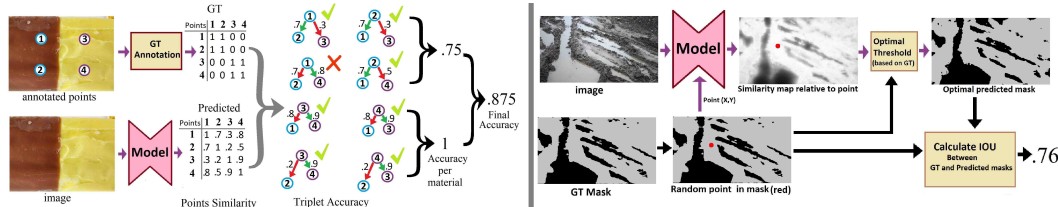

Figure 5: **Left: Evaluation on MatSeg benchmark:** 1) Predict the similarity between each pair of points in the image. 2) For every set of 3 points, choose one as an anchor and determine which of the remaining two points is more similar to the anchor according to both the prediction and GT. If both agree, mark this as correct. The average accuracy per material is the final score. **Right: Evaluation on additional benchmarks:** 1) For each GT segment mask, select a random point inside the segment. 2) Predict a similarity map showing how similar all the image pixels are to the selected point. 3) Convert the similarity map into a binary mask using an optimal threshold. The threshold is chosen to maximize the IoU (Intersection over Union) between the predicted and GT masks. 4). Calculate the IoU between the GT and the predicted mask. Repeat 1-4 with 20 random points. For SAM these are also performed with multiple input points.

similarity of each pixel in the image to the selected point (Figure 5.left). To turn this map into a hard segmentation map, we choose a threshold that maximizes the IOU between the predicted and GT masks. We use the mean IOU as the benchmark score. This is done with multiple points to obtain accurate statistics. The results are given in table 2.

# 5 Segment Anything Model (SAM) and Materialistic

**Segment Anything Model (SAM)** [34] is the current leading foundation model for image segmentation. It is designed to segment anything within an image when guided by positive input points placed inside the target segment and negative points placed outside it. Trained on a vast dataset of 11 million real images and 1 billion segments generated through a semi-manual annotation, SAM exemplifies the top performance achievable with extensive manual data annotation. The model's efficacy was evaluated using 2, 4, and 8 input points, with an equal number of positive and negative points.
**Materialistic**[53] is the latest approach for segmenting materials based on visual similarity and can be used in a zero-shot manner to find regions of the same materials in images using an input point. Similar to MatSeg, it was trained using 3D synthetic data, with one major difference: all assets used for Materialistic training data are based on existing repositories, which are mostly manually made. The results of this net offers a way to isolate the effect of the infusion method used in MatSeg compared to standard 3D synthetic data approaches.

# 6 Results

Results for the MatSeg benchmark are given in table 1 and fig. 6. The net trained on the MatSeg data achieves good accuracy and significantly outperforms both SAM and Materialistic on both hard and soft cases. It can be seen from fig. 6 that the net learned to predict highly scattered and complex patterns with soft and gradual transitions and achieves this for a wide range of domains. Hence, despite learning from synthetic data, the net manages to generalize for a wide range of real-world cases. In addition, it seems to be able to mitigate reflections and shadows fig. 6. Both Segment Anything (SAM) and Materialistic nets performed far worse on the benchmark (table 1). From fig. 6, it seems that both SAM and Materialistic search for the boundary of the material based on the boundary of the object or bulk regions and struggle with scattered shapes and soft boundaries. SAM and Materialistic also seem to be less sensitive to the material state and more focused on matching materials based on coarse-grain type (fig. 6). Both nets seem to search for bulk regions with smooth boundaries, which are likely to be more representative in their training set. SAM has failed to achieve high accuracy even after receiving 8 guiding input points (compared to 1 for other methods), suggesting that despite training on a vast amount of examples, it never encountered or learned such patterns. This seems to imply a major gap in the data collected for both SAM and Materialistic. This hypothesis is further supported by the fact that the SAM and Materialistic nets outperformed

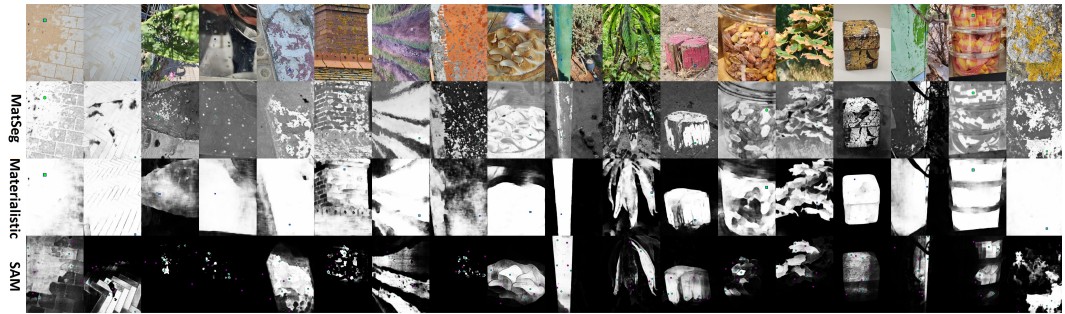

Figure 6: Results: Predicted material similarity maps to a given point in the image. The input point is marked green in each panel. For SAM net the results are with 8 input points, 4 positive (marked green) and 4 negative (marked red). The materials similarity map is brighter for high similarity and dark for low similarity (similarity to the input point). More results can be seen in fig. 12.

the MatSeg-trained net on the Materialistic test set (table 2), which contains distinct materials that encompass either a full object or distinct regions with smooth hard boundaries. As such, their shape distribution resembles that of assets used in the Materialistic train set and the SAM polygon-based annotations. Hence, the MatSeg data seems to better model raw material states, while SAM and Materialistic seem to perform better on more structured scenes and object-guided segments.

Table 1: Results on the MatSeg Benchmark (section 4). The results are in triplet loss (section 4.1). Points (pts) are the number of input points used to guide the segmentation. MatSeg **2D** and **3D** stand for nets trained on only 2D or 3D scenes, and **Mixed** stands for a net trained on a combination of 3D and 2D scenes. **All** stands for evaluation on all triplets. **Soft** stand for evaluation only on triplets with partial similarity between points(section 4).

|  | 2D MatSeg 1 point | 3D MatSeg 1 point | Mixed MatSeg 1pt | Materialsitic 1pt | SAM 1pts | SAM 2pts | SAM 4pts | SAM 8pts |
|---|---|---|---|---|---|---|---|---|
| All | 0.91 | 0.90 | **0.92** | 0.76 | 0.69 | 0.70 | 0.72 | 0.74 |
| Soft | 0.84 | 0.83 | **0.84** | 0.72 | 0.63 | 0.62 | 0.64 | 0.65 |

Table 2: Results on various of datasets (section 4.2). The results are in IOU for an optimal threshold (section 4.2). Benchmarks with **S**cattered shapes and **G**radual transitions are marked (S/G). Points (pts) is the number of input points used to guide the segmentation. The top one-shot (one point) method is **marked.** For benchmark sources see section J.

| Dataset | Field | Scatter Gradual | 1 Point MatSeg | 1 Point Materialistic | 1 Point SAM | 2 Pts SAM | 4 Pts SAM | 8 Pts SAM |
|---|---|---|---|---|---|---|---|---|
| Cu dataset | Mineral/Ore | S | **0.52** | 0.36 | 0.37 | 0.37 | 0.51 | 0.69 |
| FeM dataset | Mineral/Ore | S | **0.62** | 0.37 | 0.36 | 0.37 | 0.50 | 0.67 |
| corrosao-segment | Corrosion | SG | **0.69** | 0.49 | 0.48 | 0.48 | 0.54 | 0.56 |
| Leaf diseases | Leaf state | SG | **0.56** | 0.47 | 0.47 | 0.47 | 0.51 | 0.54 |
| URDE | Dust | G | **0.50** | 0.47 | 0.44 | 0.45 | 0.49 | 0.52 |
| Soil-type-class-2 | Soil States | G | **0.62** | 0.53 | 0.60 | 0.61 | 0.68 | 0.72 |
| Soil type | Soil types |  | 0.69 | 0.71 | **0.79** | 0.79 | 0.86 | 0.88 |
| NuInsSeg | Microscopy | SG | **0.38** | 0.17 | 0.23 | 0.23 | 0.29 | 0.32 |
| CryoNuSeg | Microscopy | SG | **0.45** | 0.28 | 0.26 | 0.26 | 0.28 | 0.28 |
| Materialistic | General |  | 0.75 | **0.87** | 0.72 | 0.72 | 0.80 | 0.85 |
| LabPics Chemistry | Chemistry |  | 0.62 | 0.63 | **0.72** | 0.72 | 0.74 | 0.75 |
| LabPics Medical | Lab/Liquids |  | **0.71** | 0.68 | 0.69 | 0.69 | 0.71 | 0.72 |

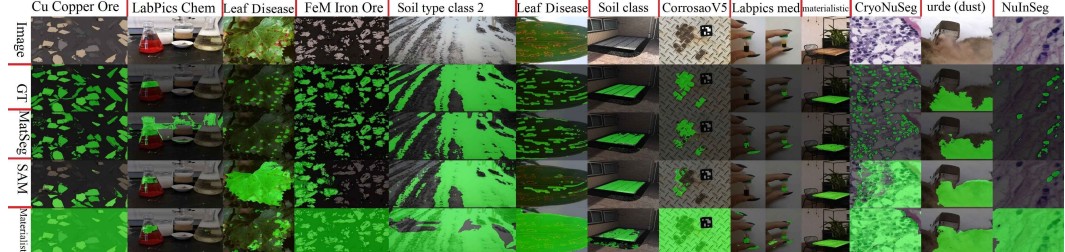

Figure 7: Sampled results on various benchmarks. The top two panels display an input image and its corresponding GT mask for a single material. The bottom three panels show the predicted masks (green) for MatSeg, SAM, and Materialistic. Predictions were generated using the image and a single random input point (four points for SAM). For benchmark sources see section J.

## 6.1 Results on other benchmarks

The results of SAM, Materialistic, and MatSeg-trained nets on various benchmarks are given in table 2 and fig. 7. These results support the conclusion of the previous section. The MatSeg-trained net outperformed other methods in cases of scattered segments, soft/gradual boundaries, and segmenting relatively similar states. Whether it is dust on roads, leaf diseases, corrosion, microscopy, or mud and dry ground. In contrast, SAM and Materialistic outperform MatSeg in benchmarks with materials that form bulk smooth boundaries, appear in single bulk shapes, have a strong correlation to objects, and don't have gradual transitions (table 2). Materialistic is more similar to SAM in its results, despite the fact that both MatSeg and Materialistic are trained on synthetic data, while SAM is trained on real images. This supports the hypothesis that the data infusion method in MatSeg taps a fundamentally different distribution that was missed by both manual annotations and synthetic data generation approaches used to train SAM and Materialistic. However, it should also be noted that the accuracy of all methods on these benchmarks is limited, suggesting that a significant gap remains in this field. Table 2 and fig. 7 show that the MatSeg net manages to generalize beyond its core task to domains like tissues and dust segmentation which are only partially defined by materials and textures. Although the accuracy in these fields was lower than in pure material benchmarks, it still outperformed SAM and Materialistic on these tasks. Note that the unsupervised texture extraction approach means that more common textures like waves or grass will be extracted more often than less common ones. This might be desirable since we want the net to be more familiar with common patterns, and even for similar patterns, no two will be the same. At the same time, this imbalance might create a deficiency of patterns in some domains. For example, MatSeg's main source of images (open-images and SAM) rarely contain microscopy images which leads to a deficiency of patterns in this domain. This might explain why the method gives lower accuracy for microscopy benchmarks.

## 6.2 Conclusion

Segmenting materials and states is fundamental to understanding nearly every aspect of the physical world and has numerous downstream applications. This work offers the first benchmark and dataset that deals with this task in a general zero-shot manner. This allows, for the first time, to train nets specifically for this task as well as to evaluate existing general methods. The results of this evaluation show that even the top foundation models struggle with this task. This work also offers a general, unsupervised approach for extracting patterns and textures from images and infusing them into synthetic scenes. This allows the creation of large-scale synthetic data that captures much of the complexity and diversity of the real world with no human effort. Nets trained on this data significantly outperformed the top foundation models on the MatSeg benchmark and many related zero-shot benchmarks (without training on the datasets). This suggests that the infusion method offers a general way to close the gap between real-world and synthetic data and tap into patterns distribution missed by leading data generation approaches. The benchmark, dataset, and 300,000 extracted textures and PBR materials have been made available (appendix A).

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

## A    Appendix: Dataset and Code Access and License

The MatSeg dataset, benchmark, full documentation, python readers, evaluation scripts, croissant metadata, and generation scripts used to create the synthetic data are permanently available at these URLs: 1, 2, 3, 4.
Over 300,000 extracted textures and SVBRDF/PBR materials, as well as the documented code used for extraction, are permanently available at these URLs: 1,2, 3, 4. Nets trained on the MatSeg Dataset, including: code, trained weights, and evaluation scripts are available at these URLs:1, 2, 3.

The dataset and code are released under the CC0 1.0 Universal (CC0 1.0) Public Domain Dedication. To the extent possible under law, the authors have dedicated all copyright and related and neighbouring rights to this dataset to the public domain worldwide. The MatSeg 2D and 3D scenes were generated using open-images dataset [35] which is licensed under the Apache License. For these components, you must comply with the terms of the Apache License. In addition, the MatSege3D dataset uses Shapenet 3D assets with GNU license.

## B    Authors Statement

We, the author(s), bear full responsibility for the content and data presented in this paper. We confirm that all data and materials used comply with relevant ethical guidelines and legal requirements. In case of any violation of rights, whether related to copyright, privacy, or any other legal claims, we assume complete responsibility.

## C    Appendix: 3D scene building

Once a map is generated (section 3.1), the 3D scene creation follows the standard methods described in previous works[21]: 1) Load random 3D objects[11, 19]. 2) Load random PBR or BSDF material for each region of the map and use the generated UV map to set the materials on the object surface. 3) Load a panoramic HDRI image to act as a background and illumination source[3]. 4) Add random objects, a ground plane, and sometimes random light sources to create shadows and occlusion in the scene. 5) Set a random camera position and render the image. Scene generation was performed in Blender 4.0[8] with the Cycles rendering engine. The generation code and the dataset have been made available (appendix A).

## D    Hardware

The 2D scene generation and materials and textures extraction were done with no special hardware on a simple CPU (i7). For the 3D dataset, a 3090 RTX graphic card was used to accelerate rendering.

## E    Appendix: Asset Sources

Images for the extraction of maps and textures were taken from the Open Images v7 dataset (Apache License[35]) and Segment Anything dataset for larger image [34]. The 3D objects for 3D scene creation were downloaded from Shapenet (GNU license)[19, 11]. 600 panoramic HDRIs for 3d scenes backgrounds and illuminations were downloaded from HDRI Haven[3] with a CC0 license. The rendering was done in Blender 4 with Cycles rendering[8]. Textureless (smooth) materials were randomly created by selecting a random value for each property (color, transmission, and reflectivity) in the Blender BSDF node.

## F    Creating PBR/SVBRDF materials by mixing

To increase the diversity of the generated PBR materials, we also mixed different PBR materials to generate new materials. This was done using previously used methods[21, 61], by taking the weighted average of the textures maps of the two materials. The mixing weights (ratios) were determined randomly either per map or per material (same weight for all properties map or a different weight for each property mixing).

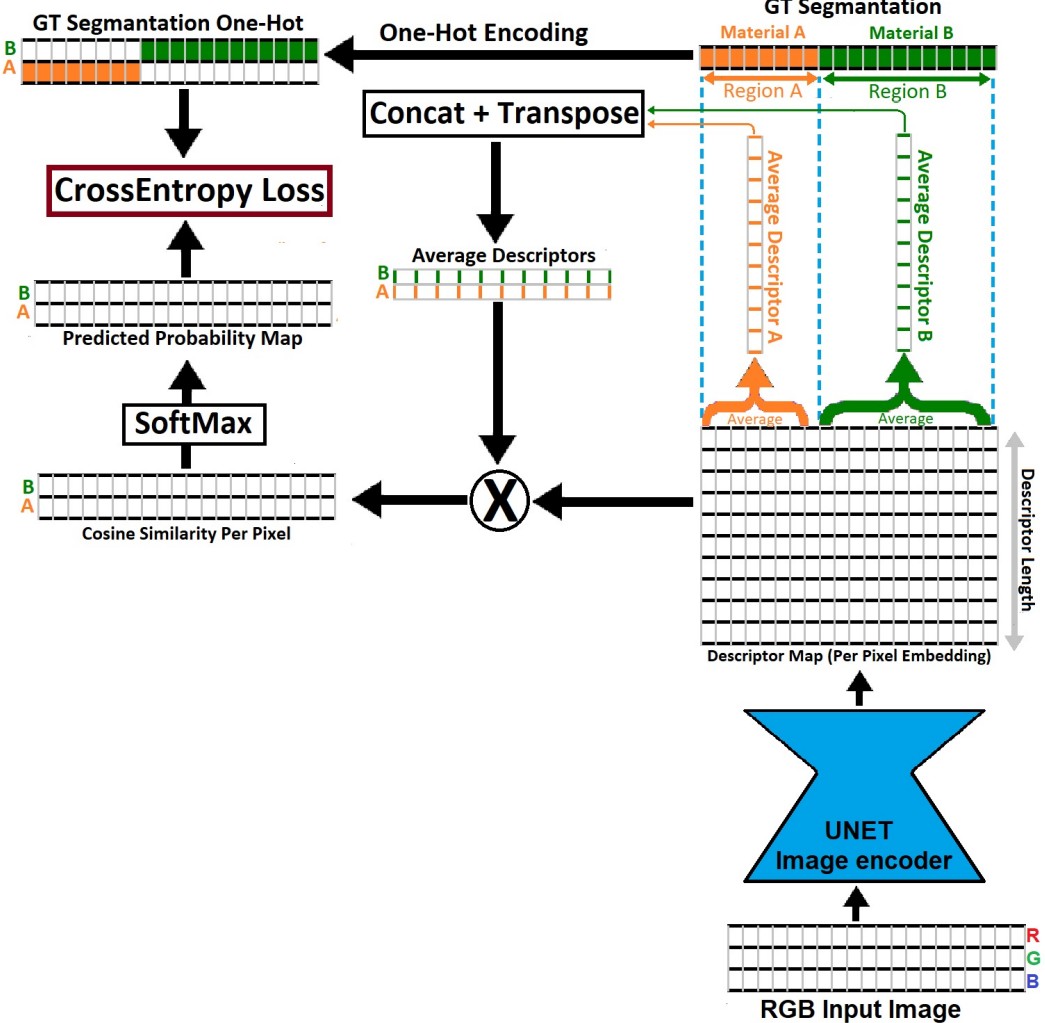

Figure 8: Net and training, a cross-section view. a) The RGB image is passed through the image encoder to create a descriptor map with a vector per pixel. This vector represents the material in the pixel. b) All the predicted vectors in the region of a given material (based on the GT map) are averaged to create the average descriptor for this material. c) Each individual vector in the predicted descriptor map is matched to the average descriptor(b) of each material in the batch using cosine similarity. This produces a 2D similarity map between each pixel and each material in the batch. d) Stacking the similarity maps and passing them through Softmax gives the predicted material probability map (predicted material per pixel). This map and the GT materials map are used to calculate the final cross-entropy loss.

# G  Net and Training

To test the MatSeg dataset, we use it to train a net for class-agnostic soft material segmentation. Previous works have shown that representing materials in images as vector embeddings and using cosine similarity between these vectors to predict how similar the materials are is an effective method for matching materials states[21]. This representation also has the advantage that it gives a soft similarity, which is good for representing mixtures, transition states, and partial similarities. To apply this approach to image segmentation, we use the Unet-style neural net, which predicts 128 long descriptors for each pixel in the image (128-layers output map). This 128 vector represents the material for each pixel (fig. 8). The cosine similarity between the vectors of two pixels in the map gives the similarity of the materials in these two pixels. The average vector in a given region (group of pixels) belonging to the same material gives the average vector embedding for this specific material.

## G.1  Training

The training procedure is depicted in fig. 8 and goes as follows: a) A training batch contains a few randomly sampled images from the MatSeg dataset. b) The images are passed through the net to produce an embedding map with a vector per pixel. These vectors are normalized using L2 normalization. c) The predicted vectors of all pixels belonging to the same material are averaged to create an average vector per material (fig. 8). To determine which pixels belong to the same material, we use the Ground True (GT) mask of each material and take all the predicted vectors in this region from the descriptor map (fig. 8). e) Each of the average material vectors in the batch is then matched with each vector embedding in the predicted descriptor map. This gives a 2D cosine similarity map between each material in the batch and each pixel in the image. f) The similarity maps for a given image are stacked together and passed through a softmax (temperature of 0.2) to obtain the probability maps for the match between each pixel and each material. g) These probability maps, along with the GT one-hot masks, are used to calculate a cross-entropy loss. Note that the GT maps are turned from soft to hard (one-hot) segments by selecting for each pixel the GT material with the highest concentration (the highest value according to the GT map).

## G.2  Net Architecture and Training Parameters

The net architecture follows that of standard U-net, with a pretrained ConvNext[38] base as an encoder, a PSP layer, and three skip connections upsampling layers (code and training scripts supplied). Training was performed using the AdamW optimizer with a learning rate of 1e-5 on a single RTX 3090. The image size was randomly cropped and resized to between 250-900 pixels per dimension. Augmentation includes darkening/lighting, Gaussian blurring, partial and full decoloring. The full code and trained models have been made available (appendix A).

# H  Benchmark annotation strategy and points selection

The goal of the point selection for the benchmark (section 4, fig. 11) was to sample the distribution of each material with an emphasis on sparse regions and transition states while focusing only on regions that the annotator can mark with a reasonable degree of certainty. This means that for scattered materials like drops on the surface, the goal will be to choose points in different drops instead of multiple points in a large drop. Similarly, for dendrite structures or twisted surfaces, the goal was to capture points that represent the surface or dendrite with point sizes that match the size of the droplet/dendrite. In the case of materials with gradual transitions, the objective is to identify points that clearly belong to distinct states, as well as points that represent the transitional state between them. Additionally, we tried to capture the "topology" of the material distribution. For instance, if two points are marked as belonging to different regions of material A, and material B exists between them (e.g., two droplets of A on a surface of B), we try to mark the point representing material B between the two points representing material A. Another objective is to capture different appearances of the same state. It is important to note that these guidelines are often hard to execute and leave a considerable amount of uncertainty. The annotation process is complex, as it is often challenging to determine whether two regions are in the same state or which points best represent the material's distribution. As a result, this process is exhaustive and heavily reliant on human judgment. To further

validate the benchmark the annotation was crosschecked by a second annotator who "corrected" errors by adding or removing controversial points.

# I   Texture extraction parameters

Textures were extracted from both open-images dataset [35] and segment anything dataset (SAM) [34]. For open images that contain smaller images on average, the tile size was limited to 40x40 pixels and a minimum of 6x6 cells per texture. For SAM dataset the cell size was set to 75x75 pixels while the grid size was limited to a minimum of 7x7 cells per texture leading to a minimal size of extracted textures of 525x525 pixels. Full extraction code has been made available at supporting material and at this URL.

# J   Additional datasets sources

URDE road dust segmentation[18], Source: 1.

LabPics chemistry: materials and phases in chemistry lab[24], Source:1,2.

LabPics medicine, liquid and phases in medical labs and hospitals[23], Source:1, 2.

Materialistic general materials[53], Source: 1.

CryoNuSeg (Tissues/Microscopy)[40], Source: 1.

FeM Iron ore microscopy dataset[17, 17, 26], Source: 1.

CU Copper ore microscopy dataset[16, 17, 26], Source: 1.

Corrosao V5 Rust/corrosion dataset[48], Source: 1.

Leaf-disease dataset[5]. Source: 1, 2.

NuInsSeg: Cells nuclei/tissue microscopy[39], 1.

Soil type class 2 Dataset: Soil/ground classification[49], Source: 1.

Soil type classification dataset (mud/dry/wet)[50], Source: 1.

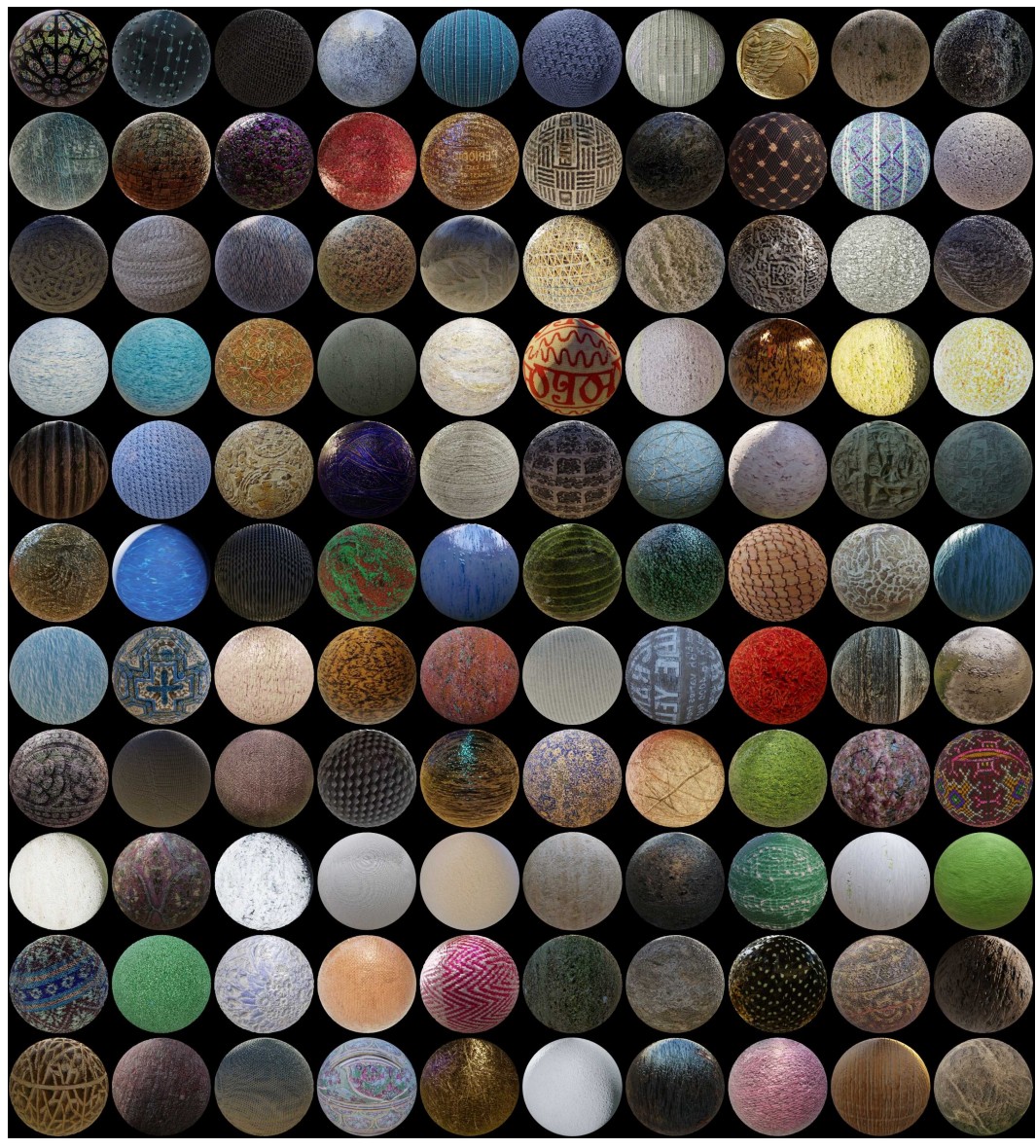

Figure 9: More samples of the over 200k extracted/generated SVBRDF/PBR materials.

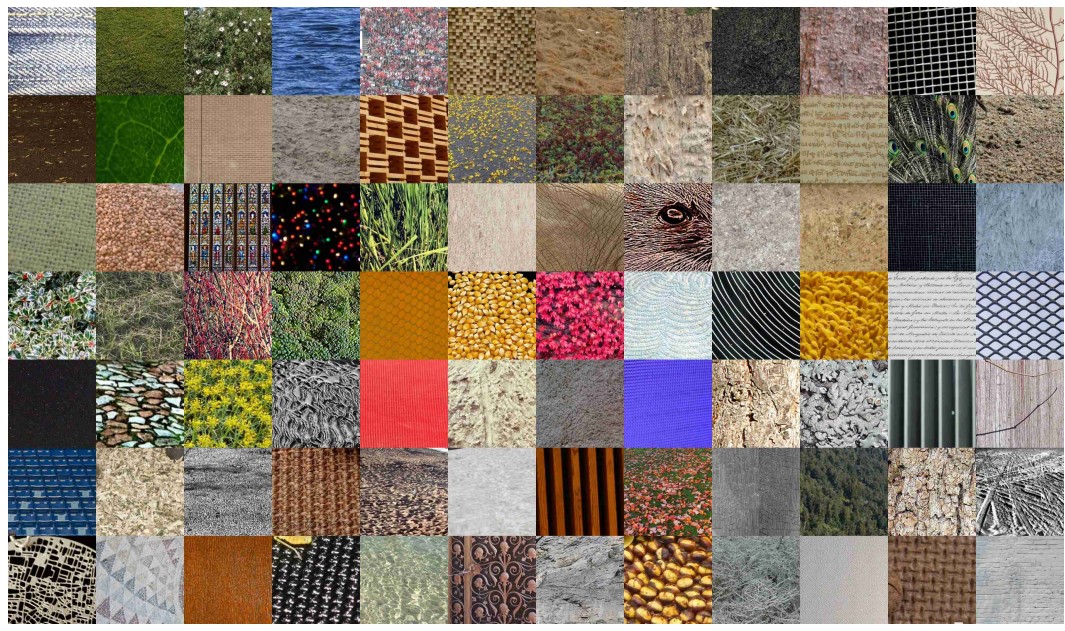

Figure 10: More samples of the over 200k extracted textures.

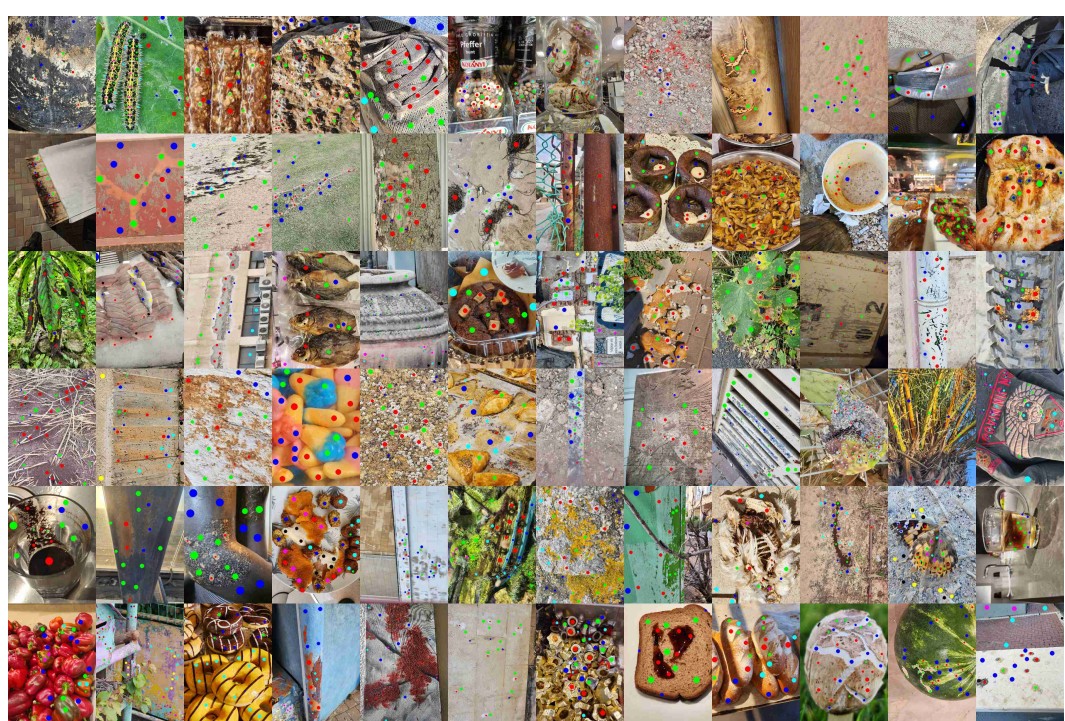

Figure 11: More examples of benchmark images with points-based annotation. Points of the same color are of the exact same material.

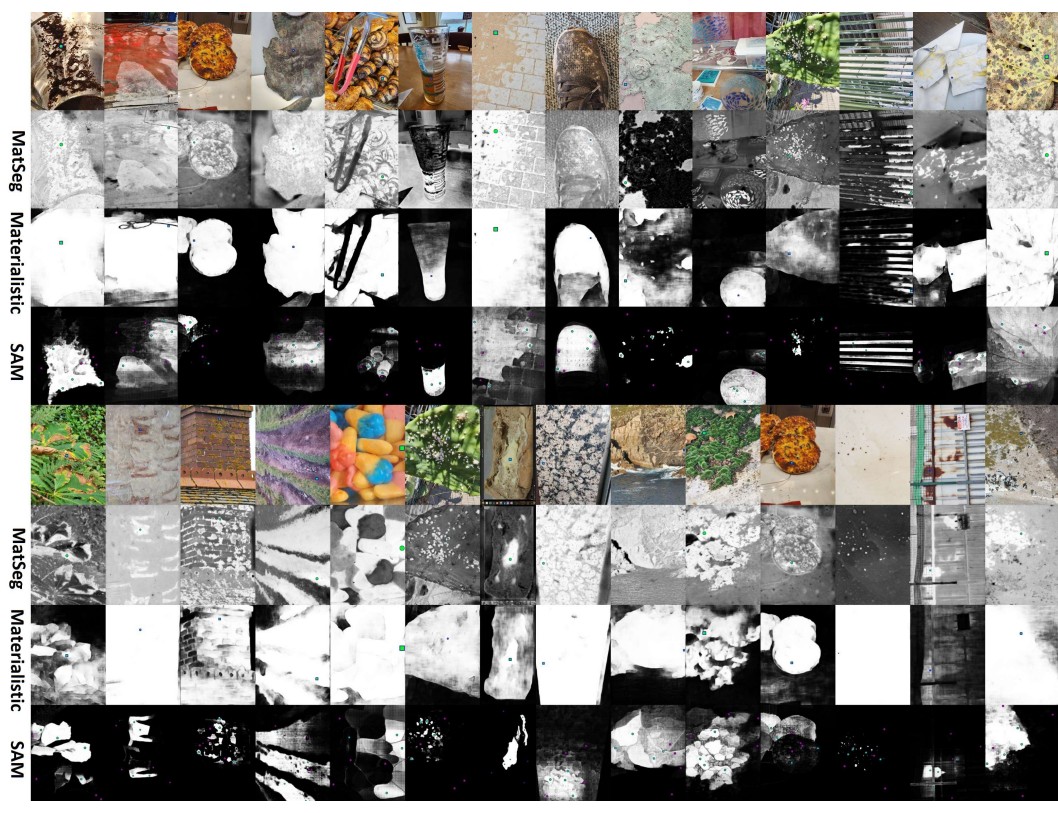

Figure 12: More examples for nets results. Predicted material similarity maps to a given point in the image. The input point is marked green in each panel. For SAM net the results are with 8 input points, 4 positive (marked green) and 4 negative (marked red). The materials similarity map is brighter for high similarity and dark for low similarity (similarity to the input point).

## 2D Senes and Annotations

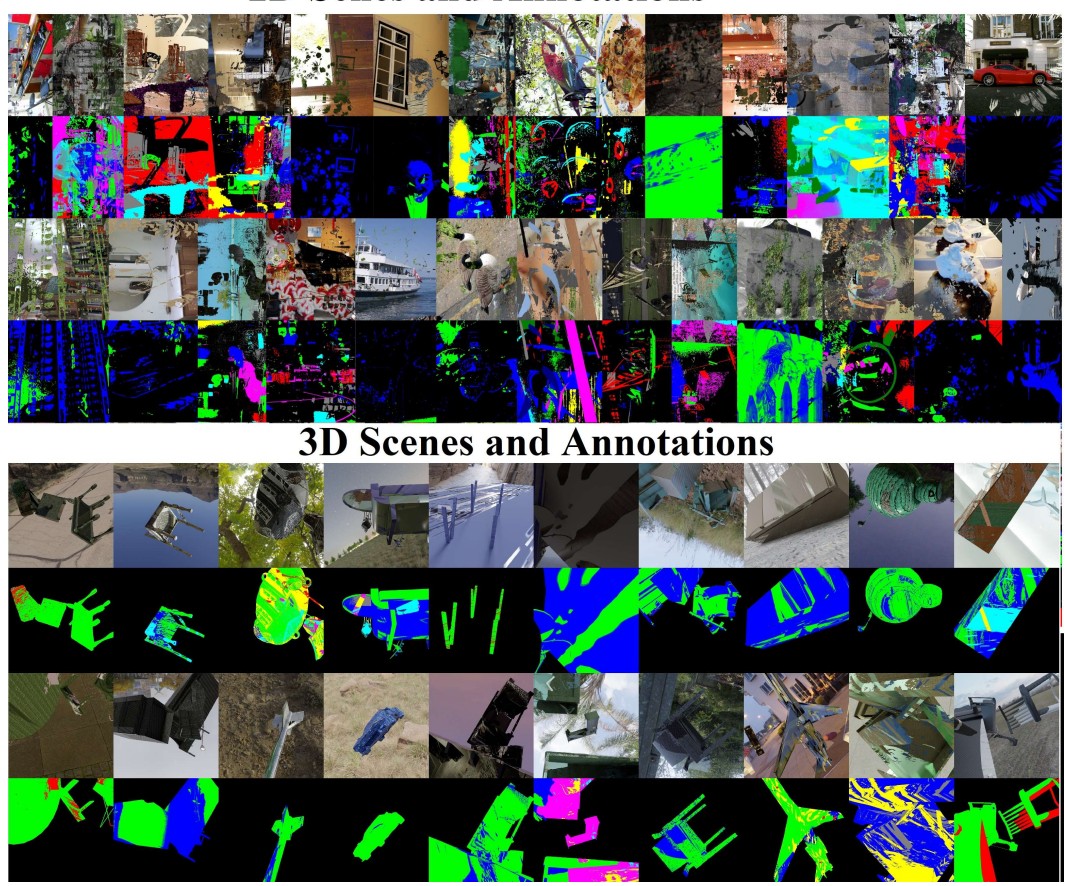

## 3D Scenes and Annotations

Figure 13: More examples of 2D and 3D synthetic scenes (top) and their annotations (bottom). In the annotation, each material is marked as a different color. A mixture of materials is marked as a weighted mixture of their colors (the weight of color is defined by each material ratio). The background is marked black.

