# OpenReview forum: "Infusing Synthetic Data with Real-World Patterns for Zero-Shot Material State Segmentation"
_NeurIPS.cc/2024/Datasets_and_Benchmarks_Track — NeurIPS 2024 Track Datasets and Benchmarks Poster_

### Official Review · Reviewer_RRue · 2024-07-22
**Overall review**

**Rating:** 6
**Confidence:** 3
**Clarity:** The paper is well organized and well …

**Review:**

1. The work is solid and well-supported by strong experimental results; see strengths.
2. There are some issues and questions; see limitations for details.

**Strengths:**

1. The proposed dataset addresses a significant gap in the field.
2. The method for pattern/texture extraction method and synthetic scene integration  strategy are simple yet effective, facilitating ease of improvement in future research.
3. The experiments and their corresponding explanations are thorough and detailed.

**Additional Feedback:**

N.A.

**Correctness:**

The dataset is constructed in a sound way. Evaluation methods in the benchmark are appropriate.

**Documentation:**

Yes.

**Ethics:**

I suspect no ethical concerns.

**Limitations:**

1. I'm curious about the fuzzy threshold selection strategy. The author mentions using random thresholds but does not provide the exact values or detailed experimental setups (e.g., random seed, number of runs, error bars for each experiment, etc.). Additionally, how does the threshold selection strategy affect the results?
2. I am also curious about the de-redundancy strategy. Could some of the extracted patterns/textures be redundant, sharing high similarity and thus being less useful for model training? If so, how is this issue addressed?

**Opportunities For Improvement:**

1. I suggest changing some figures from PNG/JPG to PDF, as the current resolution is a bit low.
2. Table 2 exceeds the text width. The author may resize it or find another way to address this issue.

**Relation To Prior Work:**

The paper discussed how this work differs from previous contributions well.

**Summary And Contributions:**

This paper introduces the first comprehensive benchmark and dataset for general material state segmentation. It also presents an unsupervised method for extracting patterns and textures from real-world images and integrating them into synthetic scenes. The paper is well-organized and clearly written. I believe this is a substantial piece of work, and the claimed contributions are both significant and valuable for advancing the field.

---

> ### Author Rebuttal · Authors · 2024-08-15
>
> **Response to reviewer RRue**
>
>
> **“I'm curious about the fuzzy threshold selection strategy. The author  mentions using random thresholds but does not provide the exact values  or detailed experimental setups (e.g., random seed, number of runs,  error bars for each experiment, etc.). Additionally, how does the  threshold selection strategy affect the results?”:**
>
> For a given map (Figure 2b), we choose two random threshold values. Everything below the lower value is set to zero,  everything above the higher set to one, and between there is a linear interpolation. This produces a soft mask that captures the image gradients and patterns, and create soft and gradual boundaries between materials.
>
> For threshold value selection, we simply pick random values (making sure there is something above and below each value). The threshold values simply define which patterns will be extracted from the image and its effect changes depending on the image (which is randomly sampled).
>
> To determine the effectiveness of this approach we simply looked at a few samples of the extracted patterns and assessed that on average they looked diverse and interesting.
>
> It is theoretically possible to go beyond this and optimize the threshold range, by recreating the dataset with various ranges and training a net on each set. However, this is likely to lead to values optimized to specific benchmarks and image repositories. In addition, rendering the full dataset takes more than a week, and doing it multiple times to optimize two values that already seem to give good patterns feels unnecessary at this point. In addition to this discussion, we will add a section with detailed values for each parameter, as well as a discussion of specific heuristics to the appendix.
>
> **“I am also curious about the de-redundancy strategy. Could some of the extracted patterns/textures be redundant”**
>
> Yes,  common patterns and textures will be extracted much more often. This might be desirable since we want the net to be more familiar with common patterns, and even for similar patterns, no two will be the same. At the same time, an imbalance of patterns might create a deficiency of net performance in some areas and redundancy in others. For example, MatSeg main source of images (open-images) doesn't contain microscopy images which lead to a deficiency of patterns in this domain
>
> However, as with other datasets, it's really hard to determine how any given pattern in the data affects the performance of the final net. As a result, giving a definitive answer here is hard. We agree that better understanding and selecting patterns can have significant effects and should be further explored in future works.
>
>
>
> All other technical comments will be integrated by default.

---

> > ### Comment · Reviewer_RRue · 2024-08-30
> >
> > Thanks for the authors’ rebuttal, most of my concerns are addressed thus I’ll keep my score.

---

### Official Review · Reviewer_WJgU · 2024-07-22

**Rating:** 7
**Confidence:** 4
**Correctness:** The method is constructed in a sound …
**Clarity:** The paper is well written.

**Review:**

Please see Strengths and Limitations.

**Strengths:**

1. This paper tries to address a significant issue, that was overlooked by most of the existing works. They make great contributions, with an automatic pipeline to generate synthetic data and a fine-grained benchmark.
2. Although the data generation pipeline is somehow, the authors claim its motivation in the introduction. Also, sufficient experiments can prove the effectiveness of the proposed data.
3. The paper is well organized and easy to follow. The authors try to demonstrate each step in the method and offer solid conclusions in experiment section.

**Additional Feedback:**

NA

**Documentation:**

There are sufficient detail on data collection and organization, availability and maintenance, and ethical and responsible use.

**Limitations:**

1. In the process of 2D scene mapping (Figure 2), How does the transformation go from (d) to (e)? Is it only by pixel overlaying? If so, would it cause some noise such as artifacts or disharmony? Meanwhile, a pre-defined threshold is set from (b) to (c). Are there any heuristic rules for this setting?
2. The benchmark seems like annotated by human. How do the authors guarantee the correctness of the samples, are there any cross-check methods to be applied?
3. Some format typos should be mitigated. Table 2 exceeds the limitation of the page. Moreover, some visualization figures should be shown in the process of evaluation metrics, which may make readers easy to understand.

**Opportunities For Improvement:**

See Limitations.

**Relation To Prior Work:**

The paper clearly discussed how this work differs from previous contributions.

**Summary And Contributions:**

1. The authors propose a large-scale synthetic dataset for general material state segmentation tasks. They also propose a benchmark to comprehensively evaluate the existing segmentation models.
2. In the construction of a synthetic dataset, the authors propose a straight but effective method, which infuses textures from real-world images with synthetic scenes.
3. Extensive experiments show some key insights. (1) General segmentation models can not handle material state segmentation tasks; (2) The dataset proposed can boost the performance of existing models.

---

> ### Author Rebuttal · Authors · 2024-08-15
>
> **Response to reviewer WJgU**
>
> **“In the process of 2D scene mapping (Figure 2), How does the transformation go from (d) to (e)? Is it only by pixel overlaying? If so, would it cause some noise such as artifacts or disharmony? Meanwhile, a pre-defined threshold is set from (b) to (c). Are there any heuristic rules for this setting?”**
>
> For every map (Figure 2b-c), we choose two random threshold values. Everything below the lower threshold is set to zero,  everything above the higher threshold is set to one, and between is a linear interpolation between zero and one. For threshold values selection, we simply pick random values (making sure there are pixels above and below each value). This produces a soft mask that captures the image gradients and patterns. The values of this map are the weight or blending ratios. For 2D this is simply the alpha  (the weight of the weighted average by which the images are merged). For 3D scenes, these values are used in the UV map.
>
> This means that the blending boundaries are often gradual. The profile of the boundaries gradient itself is dependent on the gradient of the image from which the patterns were taken. Hence,  the way in which the material blends into each other and into the scene is extracted from images. This can lead to many boundary blending profiles, some sharp, some soft, all derived from natural images. This point was missing from the paper and will be added.
>
> **“The benchmark seems to be annotated by humans. How do the authors guarantee the correctness of the samples?”**
>  The benchmark was manually annotated and cross-checked.  This is important since similarities of material states are often hard to agree upon between people. In general, annotation was done once and then checked by a second annotator who “corrected”  errors  by adding or removing controversial points.
>
> **“some visualization figures should be shown in the process of evaluation metrics:”**
>  A scheme to illustrate the evaluation processes will be added to the evaluation section.

---

> > ### Author Rebuttal · Authors · 2024-08-23
> >
> > Attached is a figure of the evaluation scheme.

---

> > > ### Comment · Reviewer_WJgU · 2024-08-30
> > > **Official Comment**
> > >
> > > Thanks for the author`s response. All my concerns have been addressed. Thus, I will keep my score.

---

### Official Review · Reviewer_LGQh · 2024-07-24
**A good dataset and sound benchmark for material state segmentation**

**Rating:** 5
**Confidence:** 3
**Clarity:** This paper is well written and easy t…

**Review:**

The overall quality of this paper is good. This paper uses an innovative method to generate a comprehensive dataset for material state segmentation and provides a benchmark. The authors clearly explained their ideas, although a concept needs more details. Experimentation is well conducted and the significance of the dataset is verified. This paper may be strengthened by explaining why a comprehensive dataset as the proposed one is needed, and by considering models trained for one specific domain.

Pros:
[Quality] The overall quality of this paper is good. The proposed dataset can provide a solid base for training new models to segment material states. Laborious work is done to manually select and annotate images for the comprehensive benchmark. The metric used is reasonable.
[Clarity] The method used to generate the synthesized images is clearly explained. The way the benchmark is set up is well explained.
[Originality] There are no such comprehensive datasets before which provide various material states of many real situations. The method used to generate the data is novel.
[Experiments] The proposed dataset and benchmark fill the blank of a comprehensive base for training and evaluating models for material states segmentation. Experiments show that even models trained on a large amount of data, like SAM, cannot handle the task of material states segmentation as well as models trained on the proposed dataset. Therefore, the significance of the proposed dataset is verified.

Cons:
[Clarity] The concept of “points” mentioned in line 81 is important to the benchmark. However, how the locations and sizes of those “points” are chosen is not mentioned.
[Presentation] This paper has not well explained why such a comprehensive dataset for zero shot material state segmentation is needed. Previous relevant works focus on segmentation of a certain scenario, like corrosion detection. This paper does not mention a scenario in which different kinds of segmentation is needed. Besides, models trained on one domain may perform better in this domain than a comprehensive model, which may limit the application of such comprehensive models. However, the experiments conducted only consider other generic models like SAM, without mentioning the performance of models designed for those specific datasets.

**Strengths:**

1. This paper designed a novel way to generate synthesized images that contain different textures, which may bring insights to future studies.
2. The proposed dataset and benchmark provide a solid base for training and evaluating models for material states segmentation.

**Additional Feedback:**

None

**Correctness:**

The dataset is rigorously constructed. The benchmark is comprehensive and solid.

**Documentation:**

Data generation is clearly explained. The dataset is publicly available. The overall documentation is good.

**Ethics:**

No such issues.

**Limitations:**

The limitations have been considered.

**Opportunities For Improvement:**

The authors may consider improvement in the following aspects:
1. Explain how the locations and sizes of the “points” mentioned in line 81 are selected in constructing the benchmark, for this selection is important to the quality of the benchmark.
2. In the introduction section, explain why a comprehensive like the one proposed is needed.
3. In the introduction section, explain why a model able to segment material states of different scenarios is needed. If possible, in section 6, compare the proposed model’s performance with dedicated models for each domain mentioned in this paper, like metal corrosion. It’s not clear whether a comprehensive model will perform better or worse than dedicated models.

**Relation To Prior Work:**

This paper provides a comprehensive analysis of relevant works.

**Summary And Contributions:**

This paper proposes a novel dataset and benchmark for material state segmentation. The proposed dataset and benchmark take into account the various kinds of textures seen in real situations, while previous works only focus on a specific domain of texture segmentation. The method used to extract textures from images and to infuse the textures into synthesized images in an unsupervised way is innovative.

---

> ### Author Rebuttal · Authors · 2024-08-15
>
> **Response to reviewer LGQh**
>
> **“Explain how the locations and sizes of the “points” mentioned in line 81 are selected in constructing the benchmark, for this selection is important to the quality of the benchmark.”**
>
> Point selection is an important part that was discussed only briefly, this discussion will be expanded:
>
> The goal of the point selection was to sample the distribution of the material with emphasis on sparse regions and transition states while focusing only on regions that the annotator can mark with a reasonable degree of certainty.
> This means that for scattered materials like drops on the surface, the goal will be to choose points in different drops instead of multiple points in a large drop. Similarly, for dendrite structures or twisted surfaces, the goal was to capture points that represent the surface or dendrite with point sizes that match the size of the droplet/dendrite.
>
> In the case of materials with gradual transitions, the objective is to identify points that clearly belong to distinct states, as well as points that represent the transitional state between them. Additionally, we tried to capture the "topology" of the material distribution. For instance, if two points are marked as belonging to different regions of material A, and material B exists between them (e.g., two droplets of A on a surface of B), we try to mark the point representing material B between the two points representing material A. Another objective is to capture different appearances of the same state.
> It is important to note that these guidelines are often hard to execute and leave a considerable amount of uncertainty. The annotation process is complex, as it is often challenging to determine whether two regions are in the same state or which points best represent the material's distribution. As a result, this process is exhaustive and heavily reliant on human judgment.
>
> **“In the introduction section, explain why a model able to segment material states of different scenarios is needed. If possible, in section 6, compare the proposed model’s performance with dedicated models for each domain mentioned in this paper, like metal corrosion. It’s not clear whether a comprehensive model will perform better or worse than dedicated models.”**
>
> Like most zero-shot methods, this approach does not aim to compete with domain-specific nets. However, there are many real-world cases in, which domain-specific nets cannot be used.
>
>
>  One example is autonomous chemistry labs that generate new materials and are therefore constantly encountering new and unfamiliar material states. This is likely true for other fields, with either uncertain domains or very wide domains. In many cases it is possible but not desirable to collect data due to effort/cost, especially for gradual and scattered segments that are very hard to mark.
>
> For this reason, the main focus of the evaluation is to see if the method outperforms other zeros shot methods and not domain-specific ones.
>
> Hence, the additional datasets (Table 2)  were aimed to evaluate the net performance compared to other zero-shot general methods, and not to domain-specific methods.
>
> This point was not mentioned in the paper and will be added.
>
> Comparing the net to domain-specific tasks is also problematic since most datasets use evaluation methods that are inconsistent with each other or the one we used.
>
> Another main issue is that domain-specific nets, the size of the training set, and the level of overlap between training and testing sets are more significant than the specific task or domain. Cases with a narrow domain where the training data is almost identical to the test set lead to very high accuracy while gaps between the training and testing can lead to low accuracy.  Diving into this means diving into the technical structure of the training and testing set of each dataset which other than being space-consuming and exhaustive have not yielded important insight into the topic of this work.
> We do have IOU results from domain-specific nets trained for corrosion, leaf disease, and materials phases (LabPics). These nets outperform MatSeg by 0.1  IOU for leaf and material phases and 0.2 IOU  for corrosion. The reason for these gaps is again very specific to the dataset structure and similarity of the training and testing sets. Exploring these gaps has yielded lots of technical points regarding the specific datasets, but nothing that is particularly meaningful for the topics of the paper. Since the goal here is not to compete with domain-specific net, we think that diving into this will be unnecessary.

---

### Official Review · Reviewer_vW41 · 2024-07-30

**Rating:** 6
**Confidence:** 3

**Review:**

- The paper addresses a significant gap in the field of computer vision by providing a new benchmark and dataset for zero-shot material state segmentation. The approach of infusing synthetic data with real-world patterns is innovative and has the potential to improve the generalizability of models.
- The paper could provide more comparative analysis with existing methods to better highlight the advantages of the proposed approach. Additionally, the generalizability of the infused data across different domains needs further validation.

**Strengths:**

The creation of a new comprehensive benchmark and dataset is a substantial contribution to the field.
The unsupervised method for pattern extraction and infusion is a novel approach that combines the benefits of synthetic and real-world data.
The release of a large repository of extracted textures and materials supports future research and dataset generation.

**Additional Feedback:**

No

**Clarity:**

The paper is well written, with a clear structure and detailed explanations of the methodology and results.

**Correctness:**

he claims made by the authors appear to be correct based on the results presented.

**Documentation:**

There is sufficient detail on data collection, organization, and availability.

**Ethics:**

There do not appear to be any immediate ethical concerns with the submission.

**Limitations:**

The authors acknowledge the limitations in their approach and discuss them in the paper.

**Opportunities For Improvement:**

The evaluation primarily focuses on the new benchmark, and more extensive testing across various domains could provide a broader understanding of the method's effectiveness.

**Relation To Prior Work:**

The paper clearly discusses how this work differs from previous contributions

**Summary And Contributions:**

The paper presents an approach to material state segmentation by combining synthetic data with real-world patterns. The authors aim to overcome the limitations of manual annotation and the lack of diversity in synthetic data. They introduce an unsupervised method to extract patterns from real-world images and infuse them into synthetic scenes, creating a comprehensive benchmark for zero-shot material state segmentation. The work includes the release of a dataset, code, and trained models to facilitate future research.

---

> ### Author Rebuttal · Authors · 2024-08-15
>
> **Response to Reviewer vW41:**
>
> **"The evaluation primarily focuses on the new benchmark, and more extensive testing across various domains could provide a broader understanding of the method's effectiveness."**
>
> While we provided results on 12 independent benchmarks (Table 2), these are all material-related and were mainly used to validate the conclusion on the main MatSeg benchmark.  We can dive more deeply into the performance of the net in each of these benchmarks.
>
> 1)  Table 2 clearly shows that the MatSeg net manages to generalize beyond its core task to domains like cell parts microscopy and dust segmentation which are only partially defined by materials/textures.    Although the accuracy in these fields was lower (Table 2) than in pure material benchmarks, it still outperformed state-of-the-art approaches like SAM and Materialistic on these tasks.
> In benchmarks directly related to material states (e.g., minerals/ores, corrosion, leaf disease), the MatSeg net significantly outperformed SAM.
> However, SAM showed superior performance in scenarios where materials correlated with objects or parts, or where more semantic scene understanding was required. For instance, SAM outperformed MatSeg in segmenting structured ground/rock/soil types when different ground types appeared only on roads and rocks of different types appeared in distinct piles (Soil type classification), or in lab materials where materials and liquids of different types were mostly located in separate vessels (LabPics chemistry). MatSeg also does show degradation in performance on microscopy benchmarks, probably because it didn't use microscopy images for the pattern extraction step. In addition, the 3d  scene generation engine (Blender) does not simulate the optics of microscopy images.
>
> In addition to this discussion, we propose adding figures that visually compare the results of different nets on various benchmarks.
>
> 2.  Regarding the application of the infusion method to areas beyond materials. We agree that the infusion method has broad applications in any area of synthetic data generation. This work demonstrates the effectiveness of this method for the material domain which is important step in validating this method. Exploring the infusion effectiveness in domains beyond materials is an important direction.  However,  this requires substantial research beyond the scope of a single paper.

---

> > ### Author Rebuttal · Authors · 2024-08-29
> >
> > Following the above discussion, attached is an image with sampled results on various of different benchmarks.

---

### Official Review · Reviewer_sTfb · 2024-07-30

**Rating:** 7
**Confidence:** 4
**Correctness:** Yes
**Clarity:** Good

**Review:**

See below

**Strengths:**

- The method of infusing real-world patterns into synthetic data bridges the gap between the precision of synthetic data and the variability of real-world data.
- The unsupervised approach reduces the need for costly and time-consuming manual annotations, making the process more scalable and efficient.

**Additional Feedback:**

see above

**Documentation:**

Yes

**Limitations:**

Yes

**Opportunities For Improvement:**

- While the unsupervised extraction process can introduce noise and unrelated patterns into the data, the paper does not provide detailed methods to ensure the complete elimination of noise in the dataset. This could potentially affect the performance and reliability of the models trained on this data.

- The primary focus of the paper is on material state segmentation. It does not explore other potential applications of the proposed method.

**Relation To Prior Work:**

Yes

**Summary And Contributions:**

This paper combines real-world patterns with synthetic data to improve material state recognition. It introduces an unsupervised method to infuse natural image patterns into synthetic scenes, enhancing precision and scalability. A benchmark for zero-shot material state segmentation is presented, revealing the limitations of current models and the effectiveness of this approach.

---

> ### Author Rebuttal · Authors · 2024-08-15
>
> **Response to Reviewer sTfb:**
>
> **"While the unsupervised extraction process can introduce noise and unrelated patterns into the data, the paper does not provide detailed methods to ensure the complete elimination of noise in the dataset. This could potentially affect the performance and reliability of the models trained on this data.”**
>
> The discussion on this topic  will be expanded.
> 1) A core hypothesis of the work is that a net trained on diverse but noisy data will learn to ignore the noisy parts while still learning the meaningful structures. This principle underpins models like GPT and CLIP, which learn from inherently noisy datasets. The performance of the MatSeg-trained network supports this assumption.
>
> 2) While the unsupervised method may extract patterns unrelated to materials (e.g., shadows), mapping these patterns onto 3D scenes using the Blender physics engine results in materials with realistic visual properties. For instance, even if a shape is extracted from shadow patterns in the image, in the 3D rendered scene, it will be used to map physically-based rendering (PBR) materials onto object surfaces. Consequently, some materials in the scene may appear in shapes that don't correlate directly with real-world materials, but their appearance and physical properties will remain realistic. This approach mitigates potentially problematic situations, such as actual shadows being misclassified as materials in the annotation. Similarly, if a uniform crowd is accidentally extracted as a texture and converted to a PBR material, it will be assigned unique physical properties (e.g., roughness, metallicity). While a material with such texture will not occur in the real world, it doesn't cause the net to learn fundamentally incorrect physical representation of materials. At worst, it may lead the net to over-generalization for unlikely real-world scenarios.
>
>
> **“The primary focus of the paper is on material state segmentation. It does not explore other potential applications.”:**
>
> 1) Material state segmentation is a very broad problem relating to nearly every aspect of the physical world. We also tested the nets on related datasets (Table 2) and will expand the discussion on these (although these datasets are mostly material-related).
> 2) We agree that the infusion method has potential applications in many areas of synthetic data generation, and exploring its effectiveness in domains beyond materials is important for future research. However, exploring these directions requires substantial research and is beyond the scope of a single paper (which has already reached its page limit).

---

> > ### Comment · Reviewer_sTfb · 2024-08-30
> >
> > I have reviewed the author’s response and the feedback from other reviewers. My concerns are mostly addressed, so I will maintain my rating.

---

### Author Rebuttal · Authors · 2024-08-15

We like to thank their reviewers for insightful comments. Their feedback has provided valuable points for improvement and discussion.
We have addressed all comments and will incorporate the responses into the manuscript.
All technical and editorial corrections will be integrated into the paper by default.
Responses to each reviewer are provided in their respective review panels.

We would like to highlight two points:

1) We are pleased that the reviewers recognize the novelty and significance of the dataset and benchmark. The primary objective of this paper is to provide a comprehensive benchmark and training dataset for zero-shot material segmentation. This task is fundamental to understanding the physical world and is crucial for various applications, including automated chemistry laboratories and any domain requiring material handling with unstructured and unpredictable properties. These scenarios often cannot be constrained by a predefined set of classes. Such a dataset is also important in situations where training data is scarce or difficult to obtain. While many excellent materials datasets exist, they typically focus on specific domains or class sets and rarely address scattered and soft boundaries. The MatSeg dataset does not aim to compete with domain-specific dataset but provides a more general solution where domain-specific training is not viable. The absence of general zero-shot datasets and benchmarks in this field has prevented the development and evaluation of new methods for this essential task. We hope the dataset will pave the way for more research in this field.

2) We appreciate the reviewers' acknowledgment of the infusion method's potential to address the domain gap between synthetic and real data. The reviewers correctly observed that while this method is broadly applicable, our current study focuses on materials state segmentation. We agree that the infusion method has potential applications in various areas of synthetic data generation, and exploring its effectiveness in domains beyond materials is an important direction. However, the paper's primary focus on materials segmentation is already extensive, requiring substantial work and testing across multiple datasets.
Validating the infusion method in the materials domain represents an important step in establishing its efficacy. Investigating this method in other fields extends beyond the scope of a single paper (which has already reached its page limit) and will be pursued in future works.

---

### Comment · Area_Chair_WXuQ · 2024-08-30
**Please Respond to Authors' Rebuttal**

Dear Reviewers,

The discussion will end soon. Please respond to authors' rebuttal and indicate whether it addressed your concerns.

Best,
AC

---

### Decision · Program_Chairs · 2024-09-26

**Decision:**

Accept (Poster)

**Comment:**

This paper presents a dataset for material recognition/segmentation in images. The research problem is somewhat interesting. The authors designed a novel dataset construction method with unsupervised pattern extracting and then infusing real world patterns into synthetic data. The construction process may inspire data synthesis in other domains.